# Influence of Front-End Electronics on Metrological Performance of QCM Systems

**DOI:** 10.3390/s24113401

**Published:** 2024-05-25

**Authors:** Ada Fort, Elia Landi, Riccardo Moretti, Marco Mugnaini, Consolatina Liguori, Vincenzo Paciello, Salvatore Dello Iacono

**Affiliations:** 1Department of Information Engineering and Mathematics, University of Siena, 53100 Siena, Italy; elia.landi@unisi.it (E.L.); riccardo.moretti@unisi.it (R.M.); marco.mugnaini@unisi.it (M.M.); 2Department of Industrial Engineering, University of Salerno, 84084 Salerno, Italy; tliguori@unisa.it (C.L.); vpaciello@unisa.it (V.P.); 3Department of Information Engineering, University of Brescia, 25123 Brescia, Italy; salvatore.delloiacono@unibs.it

**Keywords:** quartz crystal microbalance, measurement systems, signal processing

## Abstract

Quartz Crystal Microbalances (QCMs) are versatile sensors employed in various fields, from environmental monitoring to biomedical applications, owing mainly to their very high sensitivity. However, the assessment of their metrological performance, including the impact of conditioning circuits, digital processing algorithms, and working conditions, is a complex and novel area of study. The purpose of this work is to investigate and understand the measurement errors associated with different QCM measurement techniques, specifically focusing on the influence of conditioning electronic circuits. Through a tailored and novel experimental setup, two measurement architectures—a Quartz Crystal Microbalance with dissipation monitoring (QCM-D) system and an oscillator-based QCM-R system—were compared under the same mechanical load conditions. Through rigorous experimentation and signal processing techniques, the study elucidated the complexities of accurately assessing QCM parameters, especially in liquid environments and under large mechanical loads. The comparison between the two different techniques allows for highlighting the critical aspects of the measurement techniques. The experimental results were discussed and interpreted based on models allowing for a deep understanding of the measurement problems encountered with QCM-based measurement systems. The performance of the different techniques was derived, showing that while the QCM-D technique exhibited higher accuracy, the QCM-R technique offered greater precision with a simpler design. This research advances our understanding of QCM-based measurements, providing insights for designing robust measurement systems adaptable to diverse conditions, thus enhancing their effectiveness in various applications.

## 1. Introduction

Quartz Crystal Microbalances (QCMs) represent a versatile class of sensors achieved by depositing two electrical contacts onto the surfaces of thin piezoelectric crystal. These sensors have established themselves in numerous applications, spanning from environmental monitoring for detecting toxic gases to food industry quality control and biomedical research [1,2,3].

At the basis of QCMs lies their unique physical properties which make them electromechanical resonant systems with exceptional quality factors in both the mechanical (mass–spring–damper) and electrical domains, coupled by piezoelectricity. In practical terms, QCM measurements function by detecting variations in the crystal natural oscillation frequency resulting from mass, density, or viscosity alterations of a thin layer of the material in contact with one of its surfaces. Viscosity alterations can be significant, especially when the QCM is utilized to detect target substances that adhere to a functionalization layer and/or when operating in a liquid environment [4].

These variations Δf are typically described by equations like the Sauerbrey equation for mass changes Δm
(1)$$\Delta f=-2\frac{f_{s}^{2}}{A\sqrt{\mu_{q}\rho_{q}}}\Delta m,$$
where $f_{s}$ is the fundamental mode resonance frequency of the quartz, A is the area of the electrodes, $\mu_{q}$ is the shear modulus, and $\rho_{q}$ is the density of the quartz, and by the Kanazawa–Gordon equation for liquid applications
(2)$$\Delta f=f_{s}^{\frac{3}{2}}\sqrt{\frac{\eta_{l}\rho_{l}}{\pi \mu_{q}\rho_{q}}},$$
where $\eta_{l}$ and $\rho_{l}$ are the fluid viscosity and density, respectively.

The combination of these equations describes the QCM resonance frequency shifts in liquid environments. Other more complex models can be required in case the quartz is functionalized with viscoelastic films. As noted, the frequency shift is related to the mass added to the surface of the quartz, but it also depends on the mechanical properties of the added film and the surrounding fluid [5]. In this context, it is evident how monitoring other quantities related to the resonant behavior of the quartz can be beneficial, as it provides additional information about the interaction with the target substance.

This is usually attained by exploiting different measurement techniques, ranging from impedance analysis [6,7,8,9,10] to simpler and lower-cost ones such as those based on frequency measurement associated with dissipation monitoring [11,12,13,14,15,16], or on a variant of oscillator-based systems, called QCM-R, recently proposed to simultaneously obtain a measurement of the dissipative behavior of the quartz [17,18,19,20,21].

Frequency and dissipation monitoring, commonly known as Quartz Crystal Microbalance with dissipation monitoring (QCM-D), simultaneously evaluates the resonance frequency and time constant decay of the transient response offered by the QCM to a short excitation. QCM-R are feedback loops, including an automatic gain control amplifier and a frequency-selective feedback network embedding the quartz.

Each technique presents its own trade-offs in terms of cost, application specificity, accuracy, and measurement duration. Importantly, the interaction between measurement devices and quartz crystals is crucial, as front-end electronics can introduce loading effects, influencing measurement accuracy [22,23]. In fact, despite the exceptional sensitivity and resolution offered by quartz electromechanical resonators, they also exhibit significant sensitivity to front-end electronics loading and parasitic effects. This poses challenges across various measurement system topologies and methodologies, ultimately impacting the metrological specifications of QCM-based systems.

The purpose of this work is to investigate and understand, through experiments and modeling, the measurement errors associated with different QCM measurement techniques. The study focuses primarily on evaluating the metrological performance of QCM-based measurements, with specific attention to the influence of electronic circuits incorporating quartz resonators on the overall measurement quality. Through a tailored and novel experimental design, two custom measurement architectures—a QCM-R system and a QCM-D system—were compared under the same mechanical load conditions. In detail, the measurement results obtained by experiments conducted across Newtonian liquids with varying viscosities were analyzed. To compare the quartz parameters assessed from the measured signals, accurate digital signal processing techniques were exploited, granting estimation errors smaller than the errors due to the influence of the conditioning circuits. Comparative analysis against data from a commercial impedance analyzer provides further and comprehensive insights, complemented by relevant theoretical models.

## 2. The Measurement Problem

In this section, the measurement principle exploited in QCM systems is explained in detail, with a particular focus on characterizing the properties of Newtonian fluids in terms of density and viscosity. This aims to facilitate understanding of the experimental tests used in the last part of this paper to compare the metrological behavior of different measurement techniques. These specific mechanical loads were chosen because they can serve as references, given that liquids with varying and known characteristics can be easily prepared and utilized in repeated measurements. Restricting the discussion to these mechanical loads does not diminish the generality of the approach. In fact, the results obtained from the theoretical analysis are generalizable, and similar outcomes can be achieved even when considering different loads (e.g., incorporating rigid loads).

QCMs are typically made using AT-cut quartz, which behave like shear bulk resonant electromechanical systems. The electrical behavior of this type of resonators can be described, close to the resonance frequency and in the absence of mechanical loads, using the Butterworth–Van Dyke (BVD) equivalent circuit reported in Figure 1 [24].

The circuit is made up of two parallel branches, one given by the capacitance $C_{0}$ and the other given by the series of resistance $R_{q}$, capacitance $C_{q}$, and inductance $L_{q}$. $C_{0}$ depends on the dielectric properties and geometry of the quartz, $R_{q}$ describes the mechanical losses associated with the movement of the quartz and is called motional resistance, $C_{q}$ depends on the elastic behavior of the quartz, and $L_{q}$ depends on the resonator mass [25]. The equivalent impedance $Z_{q}$ given by such a circuit is described as a function of frequency f, as
(3)$$Z_{q}\left(f\right)=\frac{1}{j2\pi fC_{0}C_{q}}\frac{1-{\left(2\pi f\right)}^{2}L_{q}C_{q}+j2\pi fR_{q}C_{q}}{1-{\left(2\pi f\right)}^{2}L_{q}\frac{C_{0}C_{q}}{C_{0}+C_{q}}+j2\pi fR\frac{C_{0}C_{q}}{C_{0}+C_{q}}}.$$

The system thus described is characterized by series and parallel resonances, observable respectively at the $f_{s}$ and $f_{p}$ frequencies given by
(4)$$f_{s}=\frac{1}{2\pi \sqrt{L_{q}C_{q}}},\;\;f_{p}=\frac{1}{2\pi \sqrt{L_{q}\frac{C_{0}C_{q}}{C_{0}+C_{q}}}}.$$

From (4) it is possible to observe that the series resonance frequency $f_{s}$ only depends on the mechanical characteristics of the quartz. In the case of AT-cut quartz, $f_{s}$ typically assumes values around 5 MHz or 10 MHz. Focusing on the mechanical behavior of the quartz, the motional resistance $R_{q}$ participates in the definition of the quality factor Q of the series resonance, according to
(5)$$Q=\frac{1}{R_{q}}\sqrt{\frac{L_{q}}{C_{q}}}.$$

In the case of AT-cut quartz without mechanical load, Q takes values around 10,000.

When used for the characterization of liquids, at least one of the surfaces of the quartz is mechanically in contact with a fluid ‘half-space’. If the liquid has the characteristics of a Newtonian fluid, it is possible to electrically describe it as an impedance $Z_{l}$, given by
(6)$$Z_{l}=R_{l}+j2\pi fL_{l}=\frac{\left(1+j\right)t^{2}}{\mu_{q}^{2}Ad_{53}^{2}}\sqrt{\frac{\omega \rho_{l}\eta_{l}}{2}},$$
which is connected in series to the mechanical branch of the quartz, as described by the modified BVD circuit in Figure 2 [5]. In (6) $d_{53}$ is the 5,3 entry of the piezoelectric charge coefficient matrix (in collapsed notation).

Assuming that the QCM works close to the series resonance frequency $f_{s}$, from (6) it can be observed that in the case of Newtonian fluids the resistive component $R_{l}$ and the inductive component $L_{l}$ of the fluid impedance $Z_{l}$ are linked by the relation
(7)$$R_{l}=2\pi f_{s}L_{l}.$$

Considering the high value of the series resonance frequency $f_{s}$ for an AT-cut quartz, from (7) it is observed that the effect induced by the Newtonian fluid on the QCM consists of a notable increase in the overall resistance offered by the mechanical branch, with small variations also of the reactive part. Due to this increase in resistance, the quality factor Q of the quartz decreases drastically in applications in liquid, reaching values below 5000.

Figure 3 compares the impedance of a QCM in air and in liquid. The interaction with the Newtonian fluid produces a clear lowering of the quartz impedance magnitude peak, an increase of the value of its minimum, and a shift of the resonance conditions to lower frequency values compared to the unloaded working conditions. To derive the density and the viscosity of the fluid, two parameters synthesizing the quartz electric behavior are measured or assessed; one is related to the dissipative behavior (e.g., Q factor, minimum impedance magnitude, or real part of the series branch impedance), the other to the resonance frequency.

In this context we understand the importance of developing reliable measurement instrumentation, capable of correctly measuring the characteristics of quartz as its working condition vary. The extreme variability of the impedance offered by the QCM impacts the metrological performance of measurement devices in different manners depending on the measurement techniques they implement, producing substantial differences in the results they provide.

## 3. Measurement Systems

This section provides a brief introduction to the two front-end circuits employed in this study implementing the most common measurement techniques. Both front-end electronics were developed and realized in-house.

### 3.1. QCM-D

Within the QCM-D system, an excitation voltage with a burst-like pattern is provided to the quartz electrodes [26,27,28]. For this study, a sinusoidal burst voltage was utilized, represented by the equation
(8)$$V_{ex}\left(t\right)=V_{1}\sin\left(\omega_{ex}t\right)\mathrm{rect}\left(\frac{t}{T_{BURST}}\right).$$

Here, $V_{1}$ symbolizes the signal amplitude, $\omega_{ex}=2\pi f_{ex}$ stands for the excitation angular frequency, where $f_{ex}$ denotes the excitation frequency, and $T_{BURST}=\frac{2k\pi }{\omega_{ex}}$ delineates the burst duration, with k∈N signifying the number of sine cycles. Careful selection of $\omega_{ex}$ and $T_{BURST}$ is made to prevent unwanted excitation of QCM modes while optimizing signal amplitude. The resultant transient response of the quartz starting when the excitation vanishes (at *t = t*_0_), is captured using low-input impedance $\left(R_{L}\right)$ electronics, yielding a signal described by
(9)$$V_{o}\left(t\right)=V_{2}\exp\left(-\frac{t-t_{0}}{\tau }\right)\cos\left(2\pi f_{s}\left(t-t_{0}\right)\right)u\left(t-t_{0}\right).$$

Here, $\tau =\frac{Q}{\pi f_{s}}$ denotes the time constant of the mechanical system, and $u\left(t\right)$ signifies the Heaviside step function. A schematic depiction of the implemented front-end circuit is showcased in Figure 4.

Taking into account the BVD model illustrated in Figure 1, where the quality factor and the resonance frequency are expressed by (4) and (5), respectively, a correlation between the decay time constant and the resistance offered by the quartz is established as follows
(10)$$\tau =\frac{Q}{\pi f_{s}}=\frac{2\sqrt{L_{q}C_{q}}}{R_{q}+R_{l}+R_{L}}\sqrt{\frac{L_{q}}{C_{q}}}=\frac{2L_{q}}{R_{q}+R_{l}+R_{L}}.$$

For AT-cut quartz with a frequency of 10 MHz, $L_{q}$ remains approximately constant at some mH [5], while the motional resistance $R_{q}+R_{l}$ accounts for mechanical load effects, varying from a few Ω in air to hundreds of Ω in liquid. Consequently, the transient response as described in (9) exhibits a time constant reduced by at least one order of magnitude in in-liquid measurements. In the circuit configuration employed for this study, $R_{L}$ was set at 10 Ω.

The developed front-end electronic circuit block scheme is depicted in Figure 5; the excitation is given by a DDS which in this application is an arbitrary waveform generator. The DDS signal is applied to the rest of the circuit via an amplifier (OPA695, Texas Instruments, Dallas, TX, USA), used to adapt the output impedance of the DDS to the quartz impedance which can vary from few ohms to several hundred ohms. The multiplexer adopted to switch the quartz between the excitation and the acquisition front-end is a low impedance component (MAX393, Analog devices, Wilmington, MA, USA) and the control signal is driven by a trigger signal recovered from the excitation burst envelope.

The signal across the load resistance $R_{L}$ is amplified by a variable gain amplifier (V.G.A.) (VCA824, Texas Instruments) whose gain is set by a digital to analog converter (DAC) (16 bit) to adapt the amplified signal output dynamic to the one of the acquisition system. The value of the load resistance $R_{L}$ has been chosen to be negligible with respect to the quartz $R_{Q}+R_{l}$ in liquid applications. Common values for the series of $R_{Q}+R_{l}$ are higher than 100 Ω, thus $R_{L}$ has been chosen equal to 10 Ω. The amplifier output signal $V_{HF}$, is the signal which has been acquired and exploited in the following discussion to assess the measurement setup performance.

To adjust the gain of the V.G.A., $V_{HF}$ is mixed by exploiting an AD831 (Analog Devices) RF mixer with a local oscillator whose frequency is set near the quartz frequency and subsequently filtered, to obtain a signal in the audiofrequency band. This signal is acquired by an analog to digital converter (ADC) (16 bit, 1.25 MS/s) and dynamically processed to recover the peak output of the transient response and to find a suitable value for $V_{C}$. Moreover, the acquired signal in the audiofrequency band is also used from the control unit to perform a coarse frequency estimation to properly adjust the frequency of the excitation signal, varying the DDS frequency.

### 3.2. Oscillator-Based Circuit: QCM-R

Within oscillator-based QCM systems, the quartz crystal becomes an integral part of the feedback loop in an oscillator setup. This paper adopts a circuitry model outlined in [20], which relies on the Meacham oscillator design as illustrated in Figure 6.

For this circuit the Barkhausen conditions are:(11)$$A_{v}\left(\frac{R_{2}}{R_{1}+R_{2}}-\frac{Z_{q}\left(f_{o}\right)}{R_{f}+Z_{q}\left(f_{o}\right)}\right)=1.$$

Here, $f_{o}$ represents the self-oscillation frequency of the circuit. When the phase lag of $A_{v}$ is equal to 0°, the oscillator frequency aligns with the zero-phase frequency of the quartz itself offering an approximation of the series resonance frequency as described in (4) [20]. Moreover, by meeting this equation, the value of the motional resistance can be derived, given the resistances $R_{1}$, $R_{2}$ and $R_{f}$, and measuring $A_{v}$ as $R_{q}+R_{l}\approx |Z_{q}\left(f_{o}\right)|$.

It is important to note that in real circuits, even by selecting a large bandwidth amplifier, the non-zero phase of $A_{v}$ affects the outcomes, even if small. Furthermore, considering that the resonance frequency of an AT-cut quartz is around 10 MHz, even any parasites linked to the circuit components can influence the oscillation conditions and, consequently, the working point of the quartz. In these cases, the quartz impedance must be obtained starting from the complex version of (11)
(12)$$A_{v}\left(f_{0}\right)\left(\frac{Z_{2}\left(f_{0}\right)}{Z_{1}\left(f_{0}\right)+Z_{2}\left(f_{0}\right)}-\frac{Z_{q}\left(f_{o}\right)}{Z_{f}\left(f_{0}\right)+Z_{q}\left(f_{o}\right)}\right)=1,$$
and the estimate of the motional resistance will be given by Rq+Rl≈ReZqf0.

The front-end circuit utilized in this study incorporates a V.G.A. (VCA824, Texas Instruments). This allows for the adjustment of the gain $A_{v}$ through a feedback loop, ensuring that the oscillator poles maintain proximity to marginal stability, i.e., it ensures that the circuit oscillates generating a sine wave, satisfying (12). This strategy facilitates operation in alignment with the Barkhausen conditions regardless of variations in mechanical load by adapting the oscillator to the varying impedance of the quartz $Z_{q}\left(f_{o}\right)$.

Figure 7 represents the developed front-end electronics block scheme. In particular, the V.G.A. gain to let the oscillator work in marginal stability is set by the dichotomic algorithm described in [20]. The algorithm varies the V.G.A. gain $A_{v}$ through the control voltage $V_{G}$, exploiting a dichotomic search to find the minimum gain that allows the circuit to oscillate. As for the QCM-D front-end, the oscillator output signal $V_{HF}$ is mixed (AD831) with a local oscillator to obtain a signal in the audio frequency range. This signal is acquired by the control unit by a 16-bit ADC, 1.25 MS/s, and used to perform the dichotomic search. The V.G.A. control voltage $V_{C}$ is set by the control unit according to the algorithm and fed to the V.G.A., adopting a 16-bit DAC.

The signal $V_{HF}$ is the one acquired and exploited in the work to assess the performances of the front-end in the frequency domain, and the V.G.A. control voltage $V_{C}$ is used to recover the amplifier gain $A_{v}$, and thus the dissipative components of the quartz impedance.

### 3.3. Measurement Setup

To assess the metrological performance of the two proposed measurement systems, a complete experimental setup was implemented. This setup enables, through multiplexing, the connection of the two front-end circuits (QCM-D and QCM-R) to a QCM hosted in a static measurement chamber. By employing this configuration, measurements can be automatically repeated using both techniques without altering the mechanical load (the working conditions) of the QCM. This ensures that all repeated measurements are carried out under working conditions as stable as possible. The high frequency signals coming from the QCM-D V.G.A. and from the QCM-R oscillator output were acquired in time domain using a Textronix (Beaverton, OR, USA) MSO6 oscilloscope featuring a 12-bit A/D converter, to be later post-processed in the frequency domain. Sampling was conducted at a frequency of 125 MHz, with each acquisition window lasting 3 ms. The control voltage regulating the gain $A_{v}$ of the QCM-R front-end amplifier was acquired using a 16-bit acquisition board from National Instruments. The entire experimental system was managed through a LabVIEW 21.0 virtual instrument interface. A visual representation of the implemented experimental setup can be found in Figure 8.

The impedance of the QCM in the measurement chamber, in the presence of the test solutions was also measured by a commercial impedance analyzer (Wayne Kerr, Bognor Regis, UK, 6500B) to obtain reference measurements.

## 4. Signal Processing and Measured Parameters Extraction

Once the signal has been sampled, starting from the samples, as can be seen from the literature, two large categories can be considered for carrying out the measurements and extracting the desired information (analysis in the time domain and analysis in the frequency domain) [24]. The most efficient way to move to the frequency domain is to use the Discrete Fourier Transform. The whole signal processing to estimate the parameters of interest (frequency oscillation and decay time constant) was carried out using the LabVIEW environment. In this section, the digital processing techniques for estimating the fundamental parameters from the acquired signals are discussed. These techniques include the oscillation frequency for QCM-R measurements, as well as the resonance frequency and the decay time constant for QCM-D measurements. These techniques are described in consideration of the various measurement circuits setups that were described in the previous Section. The process of determining the oscillation frequency is the same way for both measuring circuits, and is based on Discrete Fourier Transform (DFT) [24,25].

Given the nature of the signal in the case of the QCM-D circuit (see Figure 9), before applying the algorithm for assessing the resonance frequency it is possible to apply some signal processing to increase signal-to-noise ratio, sensitivity, and frequency resolution. By its very nature, the signal decays over time but in contrast the noise is stationary. Therefore, if we carry on the recording data long after the signal has decayed, we will just measure noise and no signal. The resulting spectrum will therefore have a poor signal-to-noise ratio. Just adapting the time window length to the duration of the signal, establishing a threshold to define it based on the noise floor, will certainly improve SNR. Of course, we must not shorten the acquisition time too much or we will start to miss the information, with would result in a reduction in SNR. With a damped sine wave, the early parts of the signal are “more important” as it is here where the signal is the strongest. This effect can be exploited by deliberately multiplying the signal by a function which starts at 1 and then steadily tails away to zero. To improve frequency resolution, it is possible to change the sampling frequency (decimation) or increase the sample points (zero padding).

### 4.1. Oscillation Frequency Measurement

One of the most frequent techniques used to assess sine or damped sine frequency exploits the frequency domain analysis, which involves the determination of the location of the frequency of the spectrum maximum [25,26,27,28]. The spectrum is estimated using the Discrete Fourier Transform. The DFT does not calculate the entire spectrum but only N samples, all distant from each other by a quantity called frequency resolution, Δf, which depends on the sampling frequency and the number of processed points N. Consequently, the frequency is measured with a rough resolution. With this approach, the evaluation of the frequency is impacted by an error that depends on the spectral leakage.

Considering a generic signal, $x\left(t\right)$, sampled with a sampling frequency $f_{c}$, the DFT, $X\left(k\right),$ is evaluated on N time samples, corresponding to N spectrum samples
(13)$$X\left(k\right)=\overset{N-1}{\underset{n=0}{\sum }}w\left(n\right)x\left(n\right)e^{-jk\beta_{n}},\;\;k=0,1,\ldots ,N-1,$$
where $w\left(n\right)$ are the time samples of the used window function (if a rectangular window is used $w\left(n\right)$ = 1 for each n) [29], whereas *x*(*n*) are the samples of the signal. With a sinusoidal signal, the signal spectrum is constituted by the window spectrum located at the sinusoidal frequency $f_{x}$; see Figure 10, where the amplitude spectrum, *M*, and its DFT samples, *M*(*k*), are reported.

In order to improve the frequency estimation, it is possible to use some kind of interpolation; in the literature, the most popular types of interpolation algorithms [30,31,32] are those that are based on the knowledge of the frequency transform of the time frame that is utilized to evaluate the signal. The purpose is to evaluate the deviation of the peak position with respect to the frequency samples of signal spectrum, i.e., frequency bins. This is illustrated in Figure 10, in which the position of the spectral peak is shown together with the deviation δ of the peak position from one of the two closest frequency bins [33,34].

To evaluate the resonant frequency of the QCM, the analyzed signal was assumed to be represented by the following equation:(14)$$x\left(n\right)=A\cos\left(2\pi f_{s}nT_{c}+\phi_{0}\right)w\left(nT_{c}\right),$$
where $f_{c}$ is the sampling frequency, $T_{c}=\frac{1}{f_{c}}$ is the sampling period, and $w\left(nT_{c}\right)$ is the considered time window of length $NT_{c}$. The signal model $x\left[n\right]$ has been adopted in both considered cases presented in this work. In fact, in case the signal is a damped sine wave (as for QCM-D), (14) can be used if the window length is much shorter than the transient duration (about 5τ).

Given the signal $x\left(n\right)$ in discrete time domain, $X\left(k\right)$ represents its Digital Fourier Transform (DFT), from which the magnitude $M\left(k\right)$ can be calculated as $M\left(k\right)=\left|X\left(k\right)\right|$. When the magnitude presents a peak value on a tone frequency located at index K, the frequency $f_{s}$ of the tone can be obtained as $f_{s}=\left(K+\delta \right)\Delta f$, where $\Delta f=\frac{f_{c}}{N}$ is the DFT frequency resolution and $\delta \in \left[-\frac{1}{2},+\frac{1}{2}\right]$ is the fractional bin deviation [35,36,37,38,39,40].

To estimate the fractional bin deviation δ, different interpolation algorithms can be used, accounting for the selected window function. In detail, given a window function $w\left[n\right]$ in time domain and its spectrum $W\left[k\right]$ for a two-point interpolation algorithm [25], it is possible to define α as
(15)$$\alpha =\frac{\left|W\left(\varepsilon -\delta \right)\right|}{\left|W\left(-\delta \right)\right|}=\frac{\left|X\left(K+\varepsilon \right)\right|}{\left|X\left(K\right)\right|},$$
where $\varepsilon =\mathrm{sign}\left(X\left(K\right)-X\left(K-1\right)\right)$. When an analytical relationship exists between δ and α, it is possible to calculate the value of δ using the previous formula; as an example:(16)$$\delta_{R}=\varepsilon \frac{\alpha }{1+\alpha },\;\;\delta_{H}=\varepsilon \frac{2\alpha -1}{1+\alpha },$$
where $\delta_{R}$ is valid in the case of a rectangular window while $\delta_{H}$ for the Hanning window. In the case of other windowing functions, δ can be obtained by using approximation algorithms.

The interpolation algorithm also allows evaluation of the tone amplitude with higher accuracy:(17)$$A_{R}=M\left(K\right)\frac{2\pi \delta }{\sin\left(\pi \delta \right)},\;\;\;\;\;\;\;A_{H}=M\left(K\right)\frac{2\pi \delta }{\sin\left(\pi \delta \right)}\left(1-\delta^{2}\right).$$

### 4.2. Decay Time Constant Evaluation

The typical signal acquired from a QCM-D circuit can be assumed to be the damped sinusoidal signal represented in (9) and shown in Figure 11. In order to evaluate the decay time constant τ, an exponential fit can be performed on the time sequence of the relative maxima of the signal in the time domain, $V_{o}\left(jT_{c}\right),\;j=\left[\frac{t_{0}}{T_{C}}+\frac{k}{f_{s}T_{c}}\right]$, where $\left[\cdot \right]$ denotes the nearest integer. This time sequence is extracted from the acquired signal. To reduce the errors due to noise and to sampling, different exponential fitting algorithms can be used, among which the most used are Least Square fitting techniques. These algorithms are optimum when the superimposed noise has a Gaussian distribution. Other methods like Least Absolute Residual or Bisquare are preferable when the signal presents numerous outliers. Compared to the Least Square, these two are more robust.

The Least Square method is based on the minimization of the residue of the following equation:(18)$$\frac{1}{N}\overset{N-1}{\underset{j=0}{\sum }}\Gamma \left(V_{o}\left(j\right)\right){\left(h\left(j\right)-V_{o}\left(j\right)\right)}^{2},$$
where N is the length of the data window. The algorithm minimizes the square difference between the fitted exponential function h(j) and the maxima sequence $V_{o}\left(j\right)$, weighted using the function Γ of the j-th sample of the sequence.

## 5. Experiments and Results

The proposed algorithms were used to analyze the experimental signals obtained using the measurement system described in Section 3, and to provide estimations of the QCM parameters.

The signals were acquired during a measurement campaign in which Newtonian liquids with different characteristics were analyzed by exploiting the two QCM-based measurement techniques (QCM-D and QCM-R). Moreover, the impedance of the QCM in the measurement chamber and loaded by these liquids was measured by an impedance analyzer. During the tests in the laboratory environment, the temperature variations remained within 1 °C.

Newtonian liquids with different characteristics were prepared starting from ultrapure water ($\rho_{l}$ = 1000 kg/m^3^, $\eta_{l}$ = 0.89 mPa·s at 25 °C) and adding different quantities of anhydrous glucose, obtaining glucose concentrations in the range (0–43% *w*/*w*). Note that glucose solutions with concentration equal to 40% w/w have $\rho_{l}$ = 852 kg/m^3^ $\eta_{l}$ = 5.40 mPa·s at 20 °C [35].

Multiple measurements (at least 10) were carried out at each concentration, multiplexing the QCM-D circuit and the QCM-R circuit as described in Section 3, and then acquiring the impedance spectrum with the spectrum analyzer.

The acquired signals were subsequently analyzed following the procedures outlined in Section 4. Specifically, an initial study was used to tune the processing techniques by determining the duration of the time window used for analysis, disregarding any spurious transient signal components. Some tests were carried out to set up the algorithm parameters, e.g., the window length and type. The data presented hereafter are those obtained adopting the Hanning window, which showed the lowest standard deviation on repeated evaluation among the considered windows for all the analyzed signals.

Moreover, for the QCM-D signals, the decay time constant was evaluated by applying (18), where the analyzed window length is adapted to the transient duration by removing samples under the threshold of 50 mV (estimated as the noise floor).

At first, the proposed signal processing [30] was applied to QCM signals obtained with ultrapure water (170 μL) over 20 repeated measurements. Then, 6 test solutions corresponding to different mechanical loads were prepared by subsequently adding anhydrous glucose to the ultrapure water in doses of 5 μL (2.8% *v*/*v* 1.8% *w*/*w*) up to 30 μL (10% *w*/*w* 15% *v*/*v*). Sequential measurements on the as-prepared solutions were performed with both considered techniques, and 20 repetitions were completed. In Table 1 the mean values of the extracted parameters and the standard deviations are summarized. In the case of the QCM-D circuit, the mean decay and the standard deviations in the decay time are also reported.

The proposed techniques allow for obtaining resolutions of few hertz (some ppms), as far as the frequency estimation is concerned and lower than 1 µs for the time constant in all the measurement conditions. Therefore, they can be applied for analyzing the errors deriving from the conditioning electronics.

### 5.1. Comparison among Different Techniques

The preceding analysis underscores the capability of the proposed method to deliver measurements of identical parameters, conducted on identical test solutions within very short time intervals. These measurements exhibit sufficient accuracy and repeatability to facilitate a thorough comparison of the two measurement techniques, highlighting the specific errors associated with the load on the front-end circuit.

Central to the comparison are the estimated QCM series resonance frequency $f_{s}$ (defined in (4)) and motional resistance $R_{q}$. While resonance frequency estimates are obtained directly from the signal processing methods outlined in Section 4, deriving the motional resistance necessitates some further steps. For QCM-R measurements, the motional resistance was obtained by indirectly measuring the oscillator amplifier gain $A_{v}$ from the acquired AGC control voltage utilizing (12). Notably, the standard deviation of motional resistance measurements was found to be less than 1 Ω. Conversely, for the QCM-D-based measurements, motional resistance was derived from decay time constant using (10).

Additionally, the discussion that will follow in this Section will also encompass a comparison among the QCM parameters obtained from the QCM-D and QCM-R systems and those derived from QCM impedance spectra for each tested solution.

Although series resonance and motional resistance cannot be directly extracted from impedance spectra [5], it is known that the series resonance frequency of a QCM lies between the frequency of the impedance spectrum minimum and the impedance at the first phase zero. Similarly, the motional resistance is situated between the minimum impedance magnitude and the magnitude of impedance at the frequency of the frequency of the first phase zero. These findings are presented in the subsequent figures.

Summarizing, in all the Figures reported hereafter, the markers represent experimental results. The measurements were performed on the different solutions, then the measured signals both from QCM-R and QCM-D were acquired and post-processed, to obtain the two parameters of interest: estimation of the resonance frequency and of the motional resistance. Regarding the data experimentally obtained exploiting the impedance analyzer (providing the reference parameters), these were post-processed as well, in order to extract the 0° phase frequency and impedance module from the impedance spectra, as well as the minimum impedance module frequency.

Figure 12 reveals that all the employed techniques yielded consistent estimates of resonance frequency and motional resistance (markers). As expected, these estimates exhibited a trend of increasing resistance and decreasing frequency as the concentration of test solutions increased. However, there was also noticeable divergence in the estimates among the different techniques, with the extent of divergence escalating with the increase of the mechanical load on the quartz.

To better understand the causes of these differences between measurements, in the next subsection we compare the experimental results with the expected theoretical behavior of the different measurement techniques as the density of a Newtonian fluid applied to the QCM increases., to assess the errors. The theoretical behavior was obtained by modeling a QCM, inserted in the different specified circuits when operating with loads given by different Newtonian fluids. To this end the QCM behavior will be modeled through BVD circuit inserted in the specific front-end circuit to account for electrical loading, and the used solutions through a Newtonian fluid behavior (offering a specific mechanical load as per (6)). Therefore, we used electrical models (circuit), mechanical models (solutions) and finally coupled electromechanical models (quartz). Each of these models were tuned to account for the experimental conditions in order exactly match and explain the experimental results.

### 5.2. Experiments Validation by Theoretical Models

As anticipated in Section 2, the electrical behavior of an unloaded QCM can be described through the Butterworth–Van Dyke equivalent circuit, represented in Figure 1, given by the parallel of a capacitance $C_{0}$ with the series of a resistance $R_{q}$, of an inductance $L_{q}$, and a capacitance $C_{q}$, which corresponds to the equivalent impedance described by (3). Assuming placing a surface of the QCM in contact with a Newtonian fluid, the Butterworth–Van Dyke circuit is modified as in Figure 2, i.e., by adding a resistance $R_{l}$ and an inductance $L_{l}$ in the series branch, whose values depend on the density and viscosity of the fluid according to (6) and are connected to each other according to (7).

The previously described experiments can be referred to this theoretical model. Specifically, the behavior of the QCM in air can be modeled via the Butterworth–Van Dyke equivalent circuit in its original form, since the density and viscosity of air are negligible, and the mechanical behavior of quartz is not affected. On the other hand, the behavior of the QCM when one of its surfaces is in contact with water, or with water and glucose solutions, must be modeled through the Butterwort-Van Dyke equivalent circuit in its modified form, since water and water/glucose solutions, have characteristics comparable to those of the Newtonian fluids described by (6) and (7).

Consequently, the experimental measurements proposed in Figure 12 and discussed in the previous subsection can be related to the theoretical model of Section 2 by initially establishing the values of the components of the Butterworth–Van Dyke equivalent circuit of the quartz used for the measurements, and subsequently applying a theoretical Newtonian liquid on it via the modified Butterworth–Van Dyke circuit, evaluated for arbitrary values of $R_{l}$ and $L_{l}$ selected according to (6) and (7).

For this purpose, we started by evaluating the parameters of the Butterworth–Van Dyke model of the quartz used for the measurements in the absence of mechanical load. This evaluation was carried out by applying the fitting model described in [5] on the impedance of the QCM in air, acquired through the Wayne Kerr 6500B impedance analyzer. Referring to the BVD model in Figure 1, we established a motional resistance $R_{q}$ equal to 9 Ω, an inertial inductance $L_{q}$ equal to 6.6 mH, an elastic capacitance $C_{q}$ equal to 38.5 fF, and a parallel capacitance $C_{0}$ equal to 9.3 pF.

Subsequently, we applied a theoretical Newtonian fluid to the QCM, adding to the motional resistance $R_{q}$ a further resistance $R_{l}$, the value of which was increased from 0 to 400 Ω. Following the properties of Newtonian fluids, according to the value of $R_{l}$, we also added to the inertial inductance $L_{q}$ an inductance $L_{l}$, defined according to (7) considering the series resonance frequency equal to the value measured in air ($f_{s}$ = 9.997640 MHz). For each load scenario thus considered, we subsequently calculated the overall impedance offered by the QCM immersed in the Newtonian fluid according to the model reported in Figure 2, evaluating the trend of the frequency and the real part of the impedance in resonance, zero phase, and minimum impedance conditions.

Finally, we evaluated the overall phase shift of the oscillator due to the amplifier and the parasitic components, setting it at around -29°. Considering this result, we applied (12) to the quartz impedances calculated for the different load scenarios, evaluating the oscillation frequency and the corresponding real part of the QCM impedance expected as the viscosity of the fluid increases.

Figure 13 shows the results of this analysis of the expected behavior of the QCM (solid lines), offering a comparison with the experimental results obtained starting from the previously described measurement campaign (markers). Specifically, the curves in the figure indicate the behavior of the QCM when the density and viscosity of the Newtonian fluid varies according to the theoretical model. The blue curve shows the expected variation according to the theoretical model of the series resonance frequency and of the real part of the impedance evaluated at that frequency. The orange curve shows the expected variation according to the theoretical model of the frequency in minimum impedance conditions and of the real part of the impedance evaluated at that frequency. The yellow curve shows the expected variation according to the theoretical model of the frequency in zero phase conditions of the impedance and of the real part of the impedance evaluated at that frequency. The purple curve shows the expected variation according to the theoretical model of the frequency in conditions of oscillation of the Meacham oscillator and of the real part of the impedance evaluated at that frequency. The behavior expected based on the theoretical model is overlapped with the data coming from the experimental measurements, which are instead represented through markers. In particular, the blue markers are the average frequencies and resistances measured by the QCM-D device, the orange and yellow markers are the average frequencies and resistances measured by the impedance analyzer, and the purple markers are the frequencies and resistances averages measured through the QCM-R device.

According to what is shown in the figure, a perfect overlap between expected theoretical behavior and experimental data is appreciated, confirming the correspondence between the behavior of the tested devices with the corresponding theoretical models. This explains the measurement errors with systematic effects related to the specific measurement applied, i.e., to the specific front-end circuit. Additionally, it can be observed that the QCM-D technique appears to be much more accurate than the QCM-R technique, as the measurement of the oscillation frequency and the decay time constant of the QCM-D transient allow for obtaining the behavior of quartz in resonance conditions with a very good approximation. On the other hand, the frequency and resistance measurements provided by the QCM-R technique suffer from the phase shift introduced by the amplifier and parasitic components, leading the quartz to operate in conditions that move further and further away from resonance as the viscosity of the fluid increases. However, it is also necessary to consider the results reported in Table 1, which show that the QCM-R technique compensates for the lower accuracy with greater precision compared to the QCM-D technique. This advantage is also supported by a lower complexity in the oscillator-based instrumentation design compared to a QCM-D front-end with comparable performance. The QCM-R, in a stand-alone instrument, can be followed simply by a digital frequency-meter, and the gain of the amplifier is a low-frequency signal that can be acquired by simple hardware to derive the motional resistance. Instead, the transient response of a QCM-D must be amplified with low noise wide band amplifiers, acquired (A/D converted) and post-processed.

## 6. Conclusions

This paper thoroughly investigated the metrological performance of QCM-based measurements, focusing specifically on the influence of electronic circuits incorporating quartz resonators on measurement quality. By comparing two custom measurement architectures—a QCM-D system and an oscillator-based QCM-R system—the study shed light on the complexities involved in accurately measuring QCM parameters under varying mechanical loads.

The research provided comprehensive insights into the behavior of QCMs in different environments, particularly in liquids, where the presence of Newtonian fluids significantly impacts the resonance frequency and motional resistance of the quartz. Through meticulous experimentation and signal processing techniques, the study elucidated the challenges and issues of accurately measuring QCM parameters, highlighting the importance of reliable measurement instrumentation capable of adapting to changing conditions.

The comparison among different measurement techniques revealed consistent estimates of resonance frequency and motional resistance across various mechanical loads. However, divergence in measurements became apparent with increasing load, underscoring the complexities inherent in QCM-based measurements, particularly in liquid environments.

Furthermore, the study conducted an in-depth analysis comparing experimental results with theoretical models, demonstrating a high degree of correspondence, and reaffirming the reliability of the tested devices. Notably, while the QCM-D technique demonstrated higher accuracy in capturing quartz behavior, the QCM-R technique exhibited greater precision, albeit with lower accuracy, and offered a simpler design.

Overall, this research advances our understanding of QCM-based measurements and provides valuable insights for designing robust measurement systems capable of accurate and reliable performance across diverse applications and environmental conditions. Further exploration in this field holds promise for enhancing the capabilities and effectiveness of QCM sensors in various industries and scientific endeavors.

## Figures and Tables

**Figure 1 sensors-24-03401-f001:**
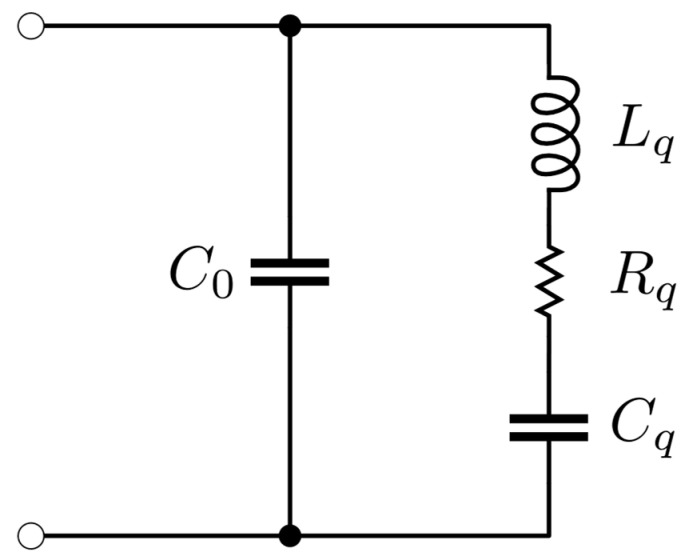
Butterworth–Van Dyke (BVD) equivalent circuit for an unloaded QCM.

**Figure 2 sensors-24-03401-f002:**
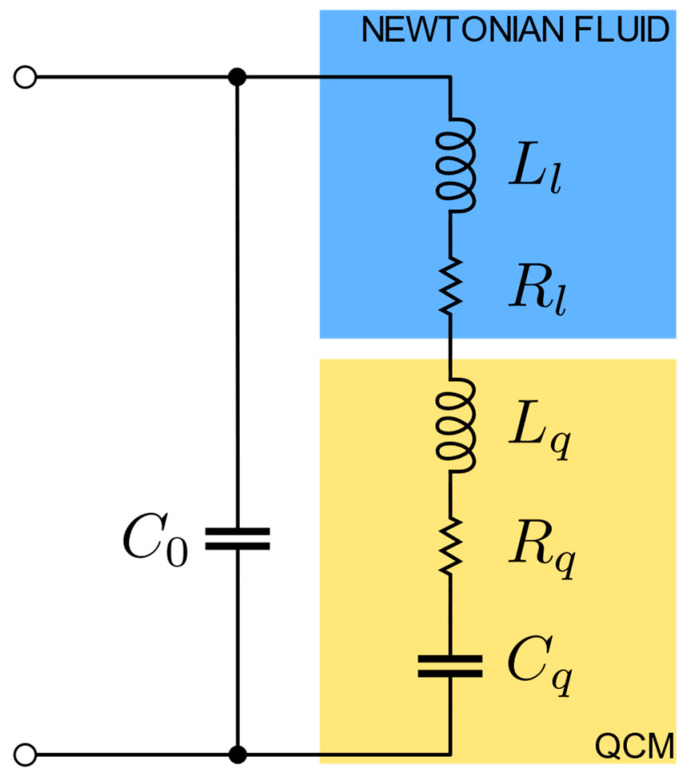
Butterworth–Van Dyke (BVD) equivalent circuit for a QCM in contact with a Newtonian fluid. The blue box highlights the Newtonian fluid impedance, the yellow box highlights the QCM impedance.

**Figure 3 sensors-24-03401-f003:**
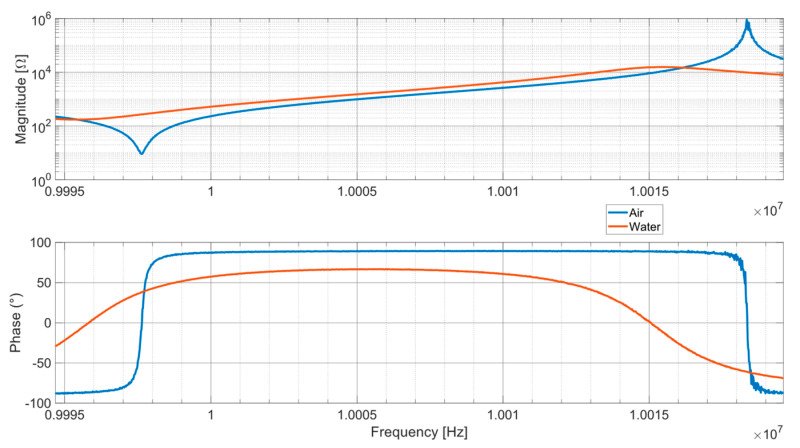
Comparison between the impedance of an unloaded QCM (blue line) and the impedance of a QCM immersed in water (orange line).

**Figure 4 sensors-24-03401-f004:**
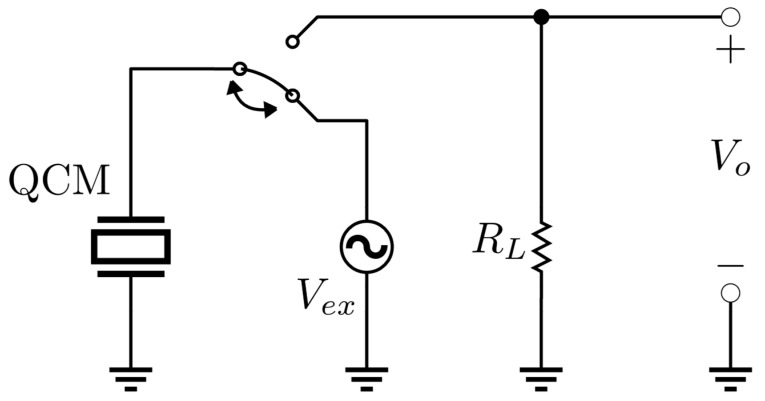
QCM-D front-end electronics circuit operating principle.

**Figure 5 sensors-24-03401-f005:**
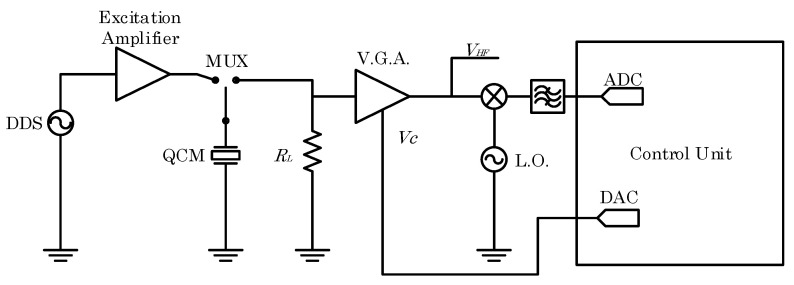
QCM-D-developed front-end electronics circuit block scheme.

**Figure 6 sensors-24-03401-f006:**
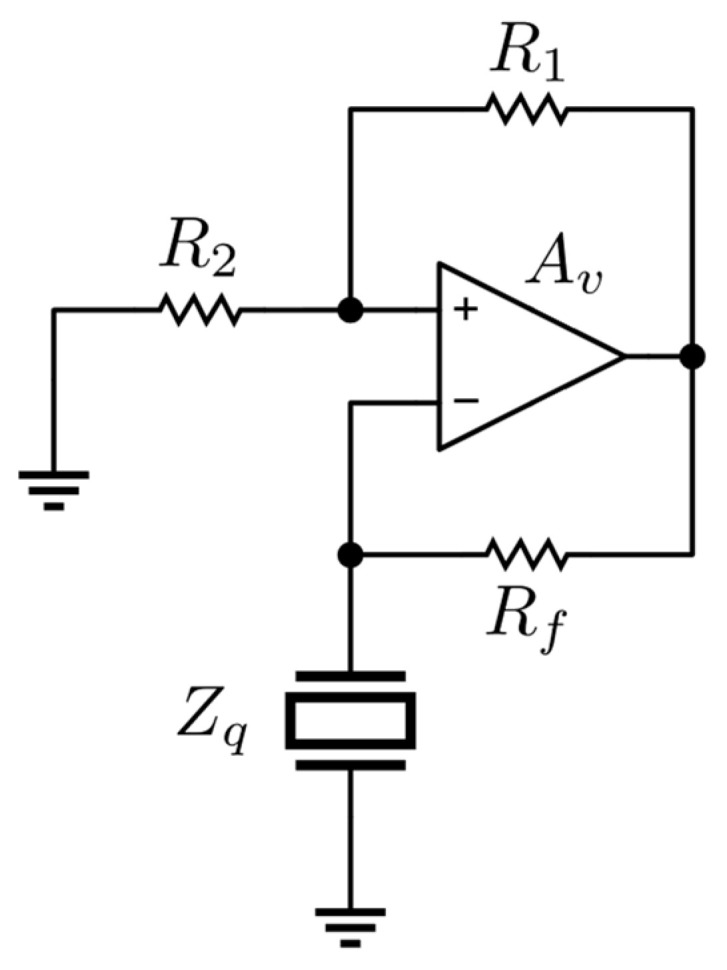
Meacham oscillator circuit topology.

**Figure 7 sensors-24-03401-f007:**
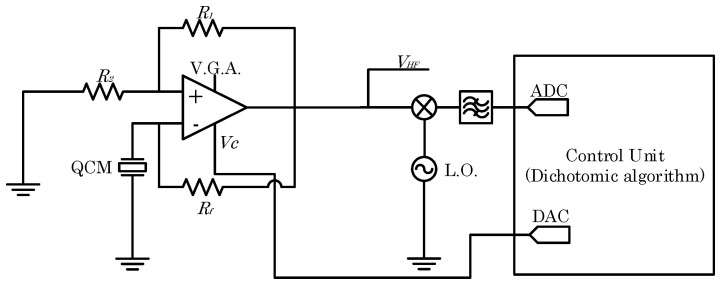
QCM-R-developed front-end electronics circuit block scheme.

**Figure 8 sensors-24-03401-f008:**
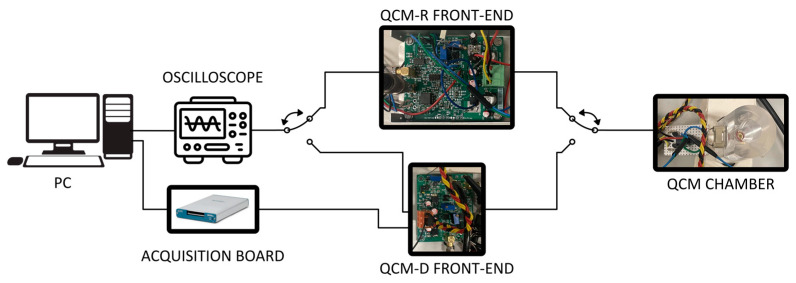
QCM measurement circuits setup block diagram.

**Figure 9 sensors-24-03401-f009:**
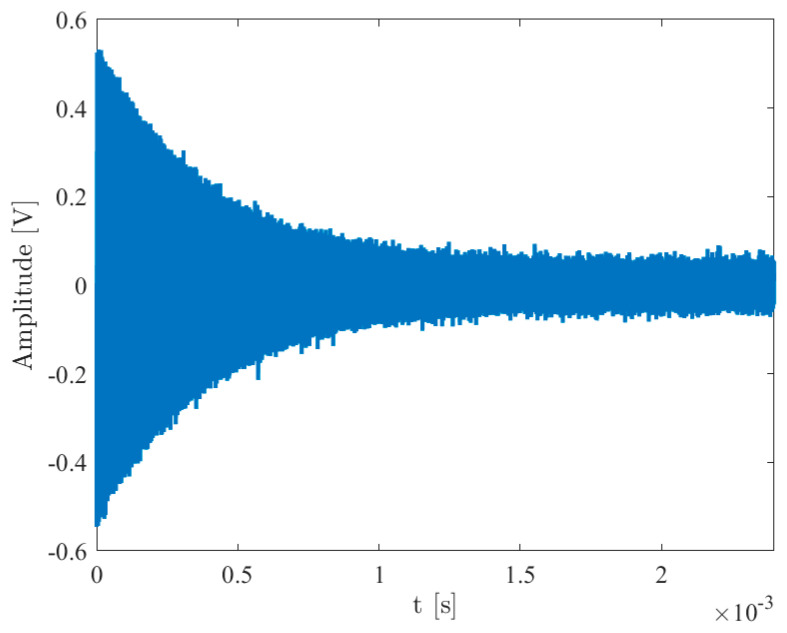
The QCM-D output signal.

**Figure 10 sensors-24-03401-f010:**
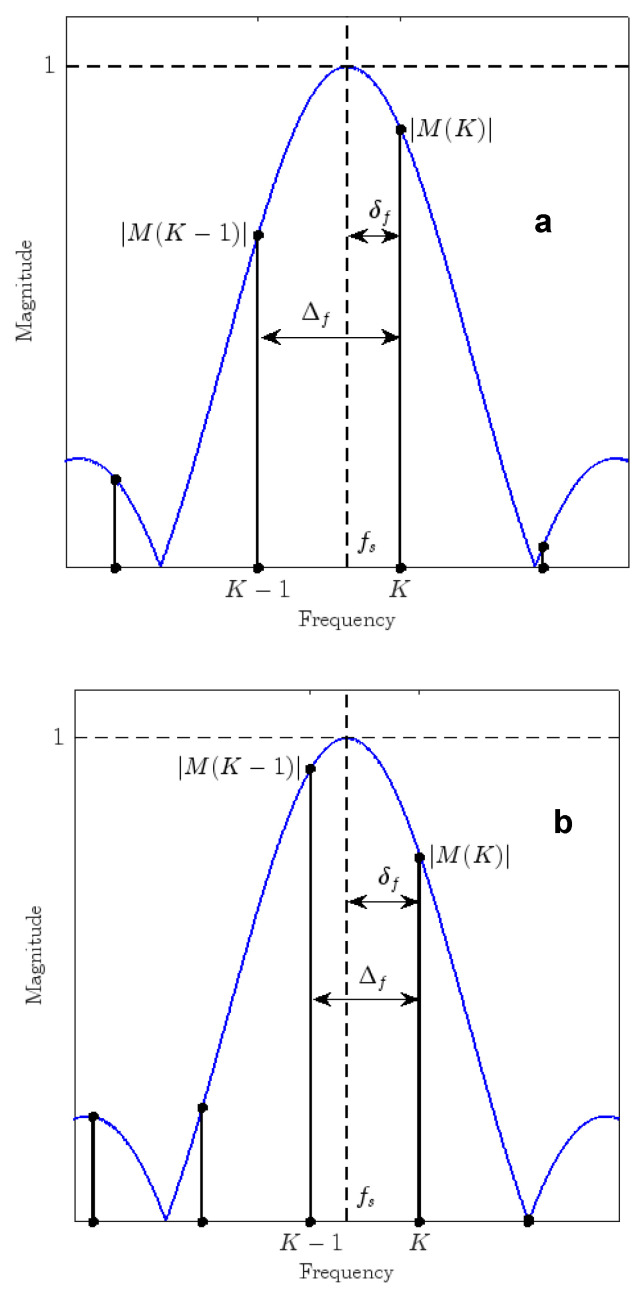
Tow tone frequency estimation problems: (**a**) Maxima is next to the M(K); (**b**) maxima is next to the M(K − 1).

**Figure 11 sensors-24-03401-f011:**
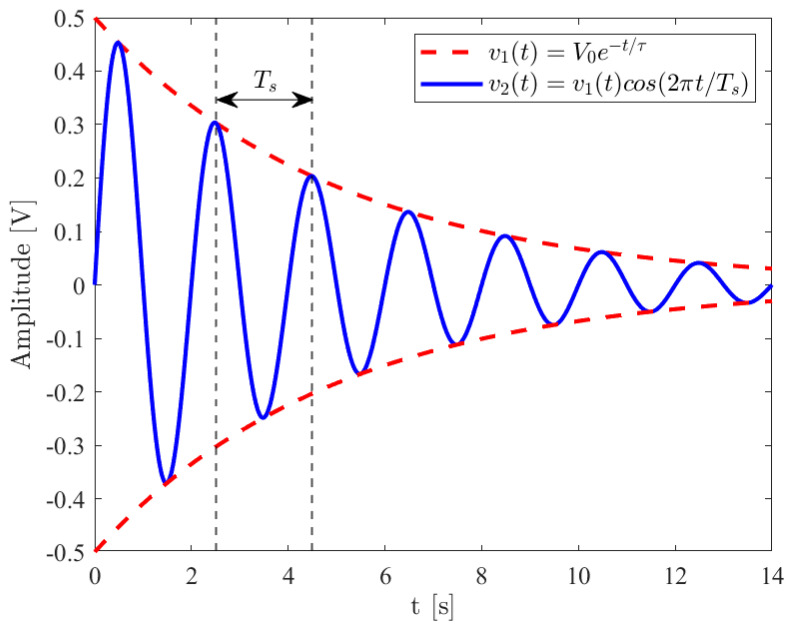
Conceptual representation of an exponentially decaying cosine wave.

**Figure 12 sensors-24-03401-f012:**
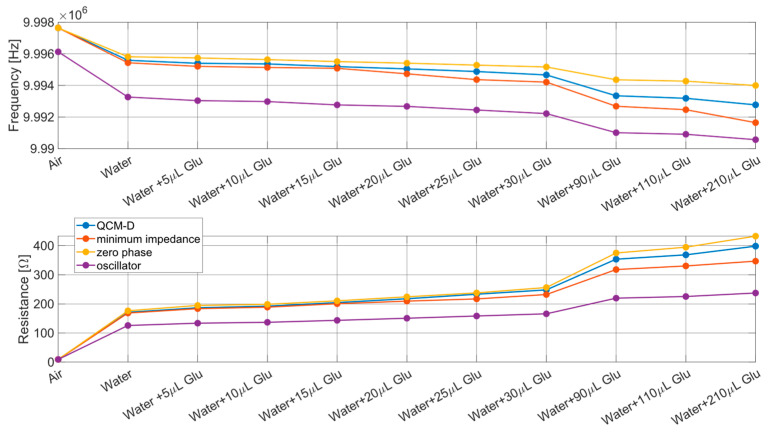
Estimates of the QCM resonance frequency and motional resistance (mean values) for different mechanical loads according to the different used techniques (blue: QCM-D, purple: oscillator, orange/yellow: impedance meter).

**Figure 13 sensors-24-03401-f013:**
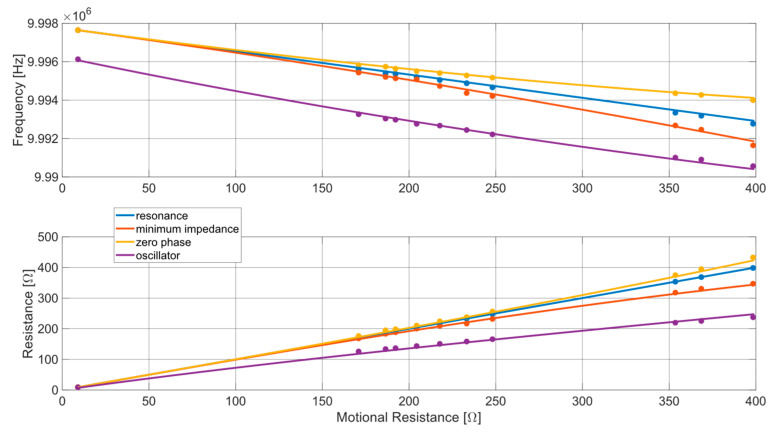
Comparison between the QCM resonance frequency (upper plot) and motional resistance (lower plot) estimations provided by the different measurement techniques, considering the experimental results (dot markers—blue: QCM-D, purple: oscillator, orange/yellow: impedance meter) and the expected behavior for a QCM immersed in a Newtonian fluid with increasing viscosity (lines—blue: resonance condition, purple: oscillator, orange: minimum impedance condition, yellow: zero phase condition).

**Table 1 sensors-24-03401-t001:** QCM-R and QCM-D systems repeatability. The mean values $\mu_{f}$, $\mu_{\tau }$ and the standard deviations $\sigma_{f}$, $\sigma_{\tau }$ of the resonance frequency and decay time constant, respectively, are reported.

Gluc [μL]	$\mu_{f}$ [Hz] QCM-R	$\sigma_{f}$ [Hz]	$\mu_{f}$ [Hz] QCM-D	$\sigma_{f}$ [Hz]	$\mu_{\tau }$ [μs]	$\sigma_{\tau }$ [μs]
0	9,993,267.1	1.8	9,995,707.4	3.4	72.53	0.22
5	9,993,042.4	4.2	9,995,563.9	3.5	66.06	0.62
10	9,992,961.2	2.6	9,995,526.4	5.0	64.49	0.33
15	9,992,770.2	2.4	9,995,408.8	5.9	60.93	0.38
20	9,992,586.0	2.9	9,995,321.4	7.6	57.07	0.44
25	9,992,403.5	2.1	9,995,197.4	7.2	53.38	0.56
30	9,992,133.4	4.7	9,995,109.4	8.7	50.27	0.67

## Data Availability

The original contributions presented in the study are included in the article.

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
