# Peer review of "Influence of Front-End Electronics on Metrological Performance of QCM Systems"

_sensors, 2024, doi:10.3390/s24113401_

Round 1
Reviewer 1 Report
Comments and Suggestions for Authors
1. For the title of the manuscript, it is suggested that the authors reflect the core innovation points, such as the core results or innovative methods;
2. In the abstract, the authors are suggested to reflect the main data to prove the innovation of the manuscript;
3. In the manuscript, many figures are not standardized, and there are even many errors. Please correct them;
4. In the part of comparative experiment, the authors are requested to mark the references of the methods being compared in order to prove the innovation of the manuscript method;
5. In the conclusion part of the manuscript, some contents still belong to the comparative experiment part, please adjust them.
Reviewer 2 Report
Comments and Suggestions for Authors
The paper starts by introducing the reader in field of QCM electronics, but after the introduction, the authors do not provide sufficient details to make the reader understand the meaning of their work and most important, their contribution.
First of all, starting from section 2, we can see that the references are not present anymore, and appear again, random, in section 3.
Second, I think the authors should clarify their contribution in the measurement setup: they designed the pcb boards (QCM-R Front-END, QCM-D Front-end) or are buyed PCB boards. More or less, they should provide some information and characteristics of the boards, being an Electronic paper (the boards appear to be very complex, but the authors specified that an amplifier with some passive components are enough (fig.5))
the term "Frequency Domain Analysis" is confusing for the readers: it is expected that the results are shown in a frequency range and normally this type of analysis is used to get the behavior of a filter circuit (for example). In fact, the authors did a transient analysis with a fourier. If they actually do a frequency domain analysis, and also measurements, than they should provide the device used for the measurements (cannot be an oscilloscope).
in the "Experimental and results " are presented only some setup information, with no experimental and results. They talked about "Signal Processing and Measured Parameters Extraction" and "Oscillation Frequency Measurement" but there is no actually measurement to prove their theory.
the authors start they paper with "The experimental results were discussed and interpreted based on theoretical models allowing for a deep understanding of the measurement problems encountered with QCM-based 23 measurement systems" , but i could not see a highlighted theoretical model, simulation results parallel with measurement results "for a deep understanding".
My recommandation for authors is to reconsider how they present their results, and in the result section to present actually results and simulation results. If they do not have something like, it should highlighted from the start the focus of the paper.
Round 2
Reviewer 2 Report
Comments and Suggestions for Authors
Even the authors tried to clarify all the things, there still are some missing information.
in their response, they said that they design the QCM-R Front-END and QCM-D Front End boards, but in low quality pictures that the authors offered, it can be seen a bar code, making me thing that the PCB boards where buyed.
So, my response will still be Reconsider after major revision until the authors offer more information about the setup: they should clarify they involvement in designing the boards, or buying the boards, the designing of SW.
Also, for this information, they describe the board. If they designed the board, they should offer a HW Architecture, and A components descriptions, etc.
